# Small hydropower plants and livelihoods of the local population in rural Vietnam

**Eva Seewald, Ulrike Grote, Trung Thanh Nguyen** *

Institute for Environmental Economics and World Trade, Leibniz University Hannover, Germany

* thanh.nguyen@iuw.uni-hannover.de

## Abstract

In the past decades, Vietnam has witnessed a great expansion of small hydropower plants (HPPs) to meet the rapidly increasing demand for electricity, improve access to water and protect against droughts. However, it is far from clear whether HPPs have exclusively positive impacts on the surrounding population as they are often accompanied by competition for natural resources (e.g., land). Therefore, this analysis investigates the impact of small HPPs on the livelihoods of the surrounding population. To this end, a unique large dataset of 2,195 rural Vietnamese households from a socioeconomic panel is combined with spatial data on HPPs. Results from fixed effects panel regressions show positive effects of HPPs on agricultural income, irrigated cropland, and the number of expected droughts. However, the effects differ depending on the location of the local population. While the downstream population benefits, the upstream population does not appear to benefit. Policymakers should, therefore, be aware of the diverse impacts in order to take effective measures and avoid social conflicts in rural Vietnam.

## 1. Introduction

To boost economic development and reduce poverty, governments around the world often invest in infrastructure [1–4] such as hydropower to generate electricity [5]. Hydropower plants (HPPs) not only provide power but also serve as a stable source of water for domestic, agricultural and industrial uses., They act as a measure to mitigate floods and droughts, meet basic human needs and improve people's quality of life while reducing greenhouse gas emissions [6,7]. However, the expansion of infrastructure intensifies competition for natural resources such as land [8–10]. As a result, it becomes more difficult for rural households to generate sufficient income from core income-generating activities such as agriculture [11]. Since the interrelationships between ecosystem and livelihood choices are immensely complex, a clear effect on the benefits or losses of the populations from infrastructure investments, i.e., the construction of HPPs, is not yet apparent [12].

While previous research has focused particularly on the effect of large HPPs on resettled communities, research on non-relocated communities in the presence of small HPPs is scarce. Large HPPs generate more than 30 MW while small HPPs generate up to 10-25 MW depending on the referred country. Medium-sized HPPs fall in the range in between [13]. In Vietnam small HPPs are defined up to a capacity of 30 MW [14]. Duflo and Pande [5] investigating the effect of large HPPs on non-relocated communities in India, state that large HPPs lead

**Data availability statement:** The data underlying the study's results are available from the

TVSEP project (www.tvsep.de) for registered users. The dataset for this paper can be downloaded from the following page http://dx.doi.org/10.13140/RG.2.2.23049.15200/1

**Funding:** German Research Foundation.

to larger irrigated land area and higher agricultural production, but also report mixed results depending on the location of the HPPs (upstream vs. downstream of villages). However, the effects of small HPPs in Vietnam on agriculture and the well-being of non-relocated communities have rarely been investigated, although most HPPs are small in size.

Therefore, it is not clear whether the effect of small HPPs is the same as for large HPPs or proportionally smaller or if the effect is completely different regarding agricultural outcomes. This is particularly relevant for Vietnam as 38% of the Vietnamese workforce was still engaged in agriculture in 2018 and around 16.3% of gross domestic product (GDP) in 2018 was generated by the agricultural sector [15]. Thus, the first aim of this study is to identify the effect of small HPPs on farming households in rural Vietnam hypothesizing that agriculture can benefit from a more stable irrigation. This, in turn, might lead to a reduction in poverty in these communities. Furthermore, the analysis of the effect of HPPs has relied primarily on case studies rather than large panel or cross-sectional data [16]. Therefore, the second aim of this study is to analyze the effect of small HPPs on the livelihoods of non-relocated rural Vietnamese households using a large panel dataset.

In order to address the two aims identified above data from a uniquely large panel dataset on rural Vietnamese households in combination with a large dataset on constructed HPPs in Vietnam is used instead of case studies for a particular HPP research [6,11,17,18]. The combination of these two data sources allows the investigation of the effect of small HPP construction on non-relocated households' agricultural outcomes and well-being. Livelihoods and well-being are investigated using different agricultural measures such as the size of cultivated land, the share of irrigated land and agricultural income, as the effect of HPPs on livelihoods should mainly be promoted due to better irrigation for agricultural purposes. Additionally, higher agricultural incomes due to better irrigation could lead to a reduction in poverty and lower income inequality. Last, the analysis of Duflo and Pande [5] is extended by examining whether the impacts differ by household location with respect to HPPs.

The results suggest that households in the vicinity of HPPs benefit by increasing agricultural income and increasing the share of irrigated land, while the effects on poverty and equality are negligible. When the effects are distinguished by location, results reveal that households downstream of HPPs benefit especially while upstream households are unaffected regarding agricultural outcomes. Therefore, the construction of HPPs is beneficial for non-relocated households. However, policymakers should implement prevention measures such as compensation schemes in order to avoid social unrest and discontent in cases where households where benefits are unevenly distributed among communities.

The remainder of this paper is structured as follows. Section 2 presents the literature and conceptual framework and Section 3 the material and methods. Section 4 presents and discusses the results while Section 5 concludes.

## 2. Literature review and conceptual framework

### 2.1. Literature review

HPPs are built to provide electricity and water for domestic, agricultural and industrial use, as well as to alleviate floods and droughts to meet basic human needs and improve the people's quality of life while preserving environmental quality [6,7]. Despite the proposed benefits of HPPs, researchers find mixed results on their impacts on local communities. HPPs have been found to result in inundation and flooding which increase salinity, toxicity of sediments and waterlogging. The loss of aquatic species and increased greenhouse gas emissions have also been associated with the construction of HPPs [19]. This in turn leads to a reduction in forests and agricultural land as well as the productivity of land which ultimately has a negative impact

on agricultural production and food security [20–28]. At the same time, the irrigated area and agricultural production increase with a higher number of HPPs in downstream areas, where farmers appear to grow more water-intensive crops [5]. Other positive effects of the construction of HPPs for the downstream population are reduced rainfall dependency, secure sources of irrigation and an increase in economic activity [5]. However, negative aspects include increased vulnerability to rainfall shocks due to restricted access to water basins, and resettlement [5,18], as well as adverse health effects (e.g., water-borne diseases) [27].

In particular, the effects on the population that had to be resettled due to the flooding caused by the construction of the HPP are being investigated. The risks for resettled populations include landlessness, unemployment, homelessness, marginalization, food insecurity, loss of access to common property resources, increased morbidity and community disintegration [28]. Most of these risks have also been verified for Vietnam [16]. Comparing resettled households to households dwelling in the village permanently the results reveal that resettled households are exposed to a loss of land and livestock resulting in lower agricultural production and on- and off-farm income [29].

Between 2009 and 2019, Vietnam recorded considerable economic growth of about 6% [30] and strong rural-urban migration [31]. Investment in infrastructure in Vietnam's urban centres was high, while rural areas lagged behind [1]. However, even rural areas in Vietnam are provided with access to electricity nearly everywhere [1], but demand for electricity continues to increase. This in turn increases investment into electricity infrastructure Nevertheless, Vietnam still faces increasing energy-cost poverty [32]. Renewable energies account for approximately 37% of energy sources in Vietnam, most of which are generated by HPPs. According to Vietnam Electricity [33], the total capacity of all hydropower resources in 2016 was 17,000 MW which is expected to increase to 27,800 MW in 2030. Expressed in numbers, this means that in 2013, 452 small-scale HPPs were either in operation or under construction [34]. Luu and Meding [18] state that there are about 7,000 HPPs in Vietnam, most of which were recently built and are small-scale in nature. However, this requires an assessment of these small HPPs as the impacts may not only be proportionately smaller than the effect of large HPPs. Results from an analysis of small HPPs in Brazil indicate that HPPs positively affect GDP, but do not affect average income, education, and life expectancy [35] while results from Ghana argue that small HPPs are useful for irrigation in semi-arid areas [36]. Additionally, Luu and Meding [18] also state that small and medium-scale HPP projects have not been sufficiently assessed regarding their impacts such as impacts on forests, aquatic life and culture. While these results hint at a beneficial effect of HPPs, the research on small HPPs investigates other climatic areas and/or builds the analysis on case studies.

Therefore, this analysis tries to close the two gaps: (i) shedding light on the effect of small HPPs instead of large HPPs, and (ii) analyzing the effect of small HPPs on the population nearby instead of resettled communities. First, the effect on agricultural outcomes is estimated. These can be used as an indicator of household well-being since HPPs should primarily affect agriculture, as shown in the literature review. Second, the impact on poverty and equality indicators is investigated. These indicators comprise the poverty headcount ratio and the Gini coefficient to measure a change in these indicators due to the construction of HPPs since previous results suggest that HPPs increase inequality [5,37] and that income inequality is a driver of energy poverty.

## 2.2. Conceptual framework

The conceptual framework is built based on Duflo and Pande [5] and Evenson and McKinsey [38]. Assuming that a rural household is equipped with a cropland area (A) and a specific amount of household labor (L), which can be used for on-farm and/or off-farm activities

denoted as $L_{farm}$ and $L_{off-farm}$, respectively. Crop production is subject to agricultural shocks (S), e.g., floods. Small farmers are assumed to be price takers with regard to farm inputs and output [39]. The construction and operation of small HPPs offer few off-farm employment opportunities. The livelihood and welfare benefits of small HPPs for small farmers are therefore mainly derived from crop production, in the form of (i) productivity-enhancing effects, and (ii) expansion of cropland [5]. These two effects are due to the availability of water for irrigation and the mitigation of agricultural shocks such as floods or droughts [6,7]. In addition, if more land can be cultivated due to the availability of water, households could also allocate more labor to crop production [5]. All those effects increase or decrease depending on the distance (from the household's cropland plots) to the HPP ($Dist_{HPP}$). Therefore, it is hypothesized that the shorter the distance, the higher the household agricultural/crop income. In addition, the effects of an HPP may also depend on where the household lives in relation to the HPP. If the HPP is located downstream of the village, households upstream may lose cropland due to inundation. Therefore, the cultivated area in upstream villages is expected to remain unchanged or even decrease, although the share of irrigated land increases. For downstream villages, the opposite effect is expected as irrigation may be lower than before the HPP and, thus, more land is needed to produce the same output. The household's income maximization problem is defined as follows:

$$\max y = f\left(L_{farm},\ L, A,\ S,\ Dist_{HPP};\ HPP_{Downstream},\ HPP_{Upstream}\right) \tag{1}$$

$$subject\ to\ L_{farm} + L_{off-farm} \leq L \tag{1a}$$

$$L_{farm} \geq 0;\ L_{off-farm} \geq 0 \tag{1b}$$

equation 1a specifies the condition that the sum of labor allocated to on-farm and off-farm activities is less than or equal to household labor, while equation 1b indicates the condition of non-negativity.

If agricultural income increases, this would help to reduce the poverty rate. In addition, as the literature shows, the rural poor are more dependent on agriculture for their livelihoods. It is therefore to be expected that an increase in agricultural income will reduce income inequality among rural households because the poor might benefit more than the better off. The mechanisms through which an HPP impacts the livelihoods and well-being of farm households are summarized in Fig 1.

## 3. Materials and methods

### 3.1. Data

Two data sources are used for the analysis. The first source is the socioeconomic data at the household and village levels from the Thailand Vietnam Socio-Economic Panel (TVSEP) project. They are used to measure agricultural outcomes such as annual agricultural income and cultivated and irrigated area per household. Demographic information for each household was also obtained from the TVSEP project. The second source is the data on hydropower from the Mekong Region Futures Institute (MRFI). The two datasets were merged using Global Positioning System (GPS) data for the villages covered by the TVSEP project and the HPPs covered by the MRFI in ArcGIS. An analysis in ArcGIS was performed to calculate the distance (in km) between villages and HPPs. Furthermore, it was also used to retrieve

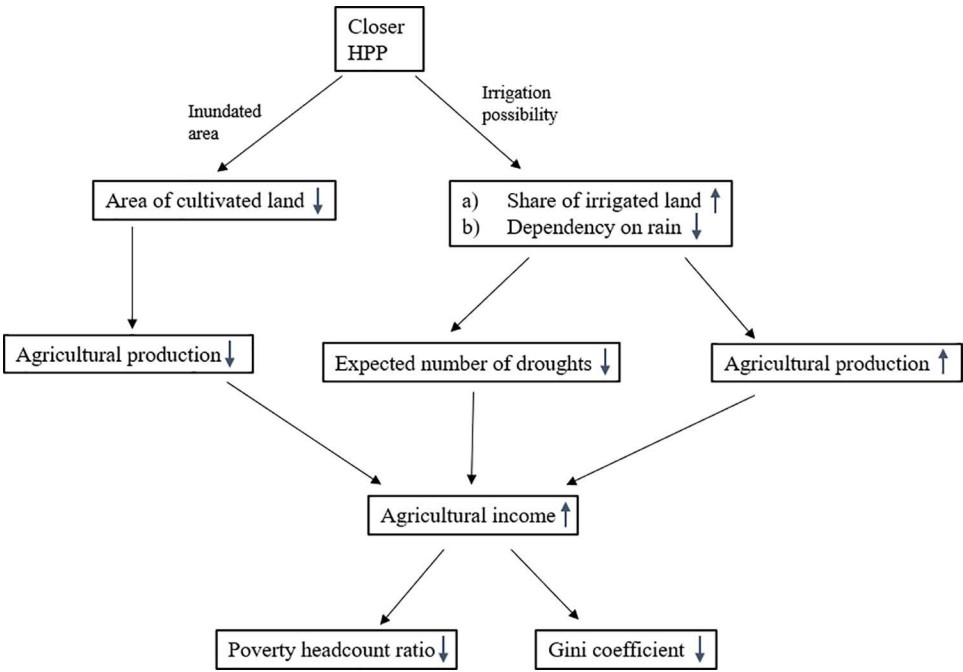

**Fig 1. Representation of channels through which HPPs affect well-being and agricultural outcomes (source: own representation).**

information on the distance and number of HPPs located either upstream or downstream of each village. The results from ArcGIS were transferred to Stata to analyze the effect of HPPs on agricultural outcomes.

The TVSEP dataset was collected under the research project "*Poverty Dynamics and Sustainable Development: A long-term Panel Project in Thailand and Vietnam* (www.tvsep.de)". This project aims to establish a long-term panel that "*enables and carries out research on long-term welfare dynamics, rural-urban migration, agricultural transformation and intergenerational aspects of households as well as on the long-term impacts of development interventions*" [40]. Important sections of the questionnaire comprise information on land, agriculture, and natural resources as they can either form a source of income or a cause of vulnerability [41]. The sampling procedure of the TVSEP was based on the guidelines of the Department of Economic and Social Affairs of the United Nations. In Vietnam, TVSEP initially selected three provinces (Ha Tinh, Thua Thien Hue, and Dak Lak) as study sites (see Fig 2). The selection of these provinces was based on criteria for assessing vulnerability to poverty which consist of four key aspects: low per capita income, high dependence on agricultural production, presence of risk factors and poor infrastructure. In the next step, a three-phase stratified random procedure was employed to sample villages and households. In the first phase, communes were selected according to the proportion of their population in sampled districts. In the second phase, two villages of each commune were selected with a probability proportional to their population size in the commune. In the third phase, ten households in each village were randomly selected with equal probability from a list of all households. This procedure resulted in a total sample of 220 villages and 2,197 households.

For the analysis, the following agricultural variables are calculated for each household: annual agricultural income, cultivated land, share of irrigated land, agricultural shocks (including flooding, drought, unusually heavy rainfall, crop and storage pests, livestock

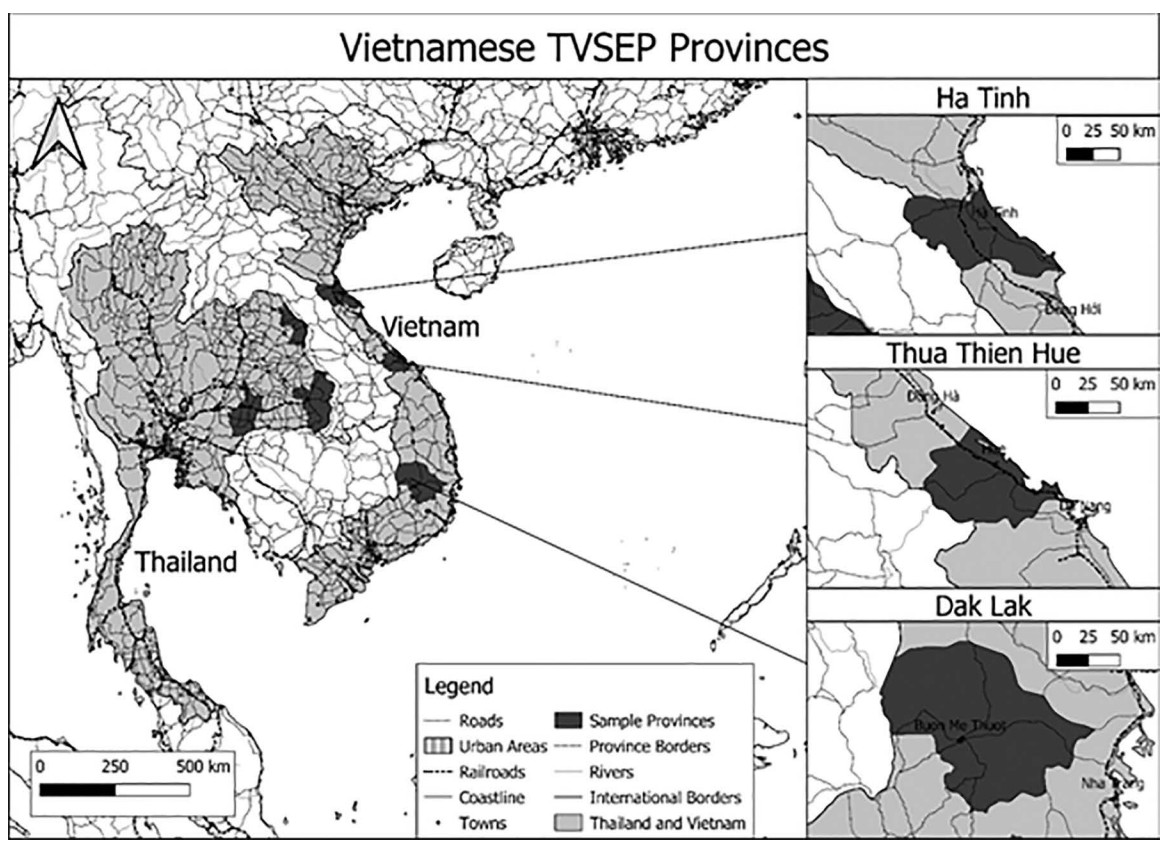

**Fig 2. Map of the Vietnamese provinces covered by the TVSEP project (source: reprinted from the Humanitarian Data Exchange (HDX) under a CC BY license, with permission from HDX, original copyright OCHA Regional Office for Asia and the Pacific (2024), own presentation in QGIS).**

diseases, and landslide/erosion), number of expected droughts, and the numbers of household members working on farm and off-farm. The number of agricultural shocks and the expected number of droughts were measured by asking the households for each kind of shock if they think it will occur and how often they think these shocks will occur in the future. More information on the definition and measurement of these variables is in the S1 Table. Additionally, socio-demographic information was extracted from the TVSEP project. This includes information of the household head such as gender, age, years of schooling, and a dummy indicating whether a household belongs to an ethnic minority. Furthermore, with respect to poverty and inequality, data from the TVSEP project was used to calculate village-wise poverty headcount ratios using the poverty line of 1.90 USD 2005 PPP and the Gini coefficients using households' annual total net income (in 2005 USD PPP).

The data on HPPs in Vietnam were provided by the Mekong Region Futures Initiative (MRFI) (formerly the Research Program on Water, Land and Ecosystems). The dataset covers HPPs with a minimum capacity of 15 megawatts and/or 0.5 km² reservoir area and was continuously maintained until the project ended. The version used in this study was last updated in June 2019. In addition, the dataset also provides GPS data and information on commission operating dates. Satellite images were used in case of missing commission operation dates to obtain information about the year in which a particular HPP started operation. The GPS information from both data sources provides an overview of the location of villages and HPPs

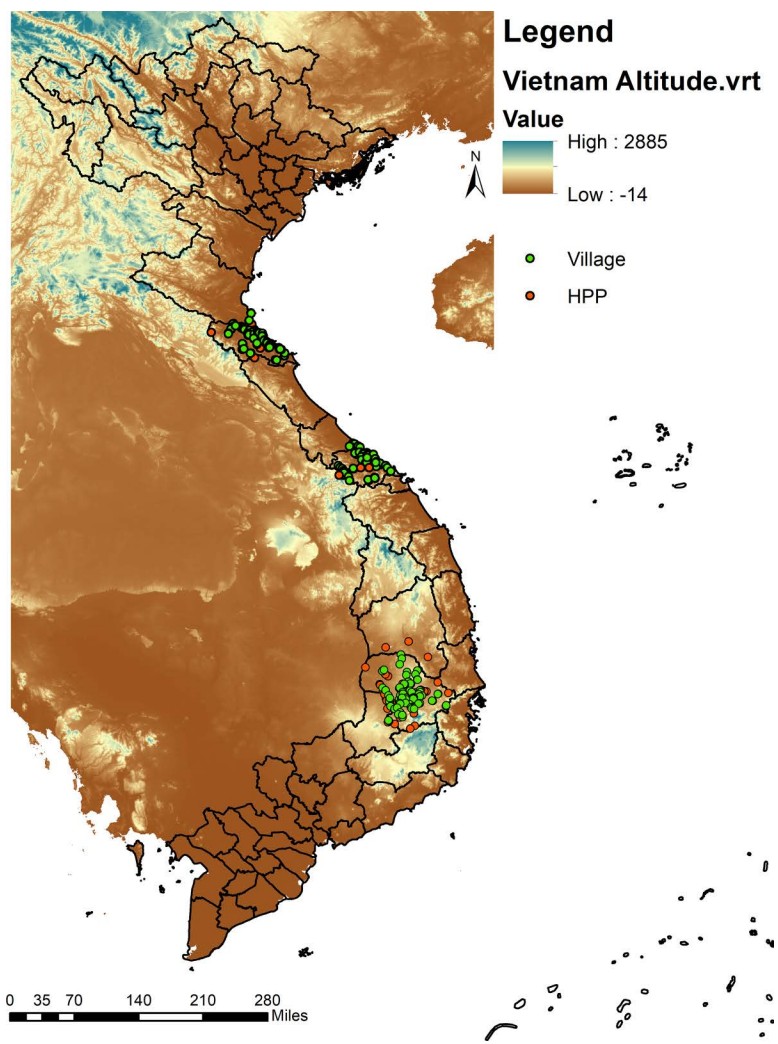

**Fig 3. HPPs' and villages' location (green dots representing villages from the TVSEP project, red dots representing HPPs) (source: own calculation in ArcGIS based on data from MRFI, HDX, and TVSEP, reprinted from HDX under a CC BY license, with permission from HDX, original copyright OCHA Regional Office for Asia and the Pacific (2024)).**

in the three provinces (Fig 3). The MRFI data shows that the highest density of HPPs within the three TVSEP provinces is found in Dak Lak.

The distances between villages and HPPs in each survey year of the TVSEP project were calculated in ArcGIS based on GPS data in order to investigate the impact of HPPs on agricultural outcomes, poverty and income inequality in rural Vietnam. Data quality during the merging process was ensured by using the same coordinate system (GCS_WGS_1984). GPS data usually contains information of six digits. The different distances from each village to the nearest HPP are due to the construction of new HPPs closer to the TVSEP villages. In addition to the distance to the nearest HPP overall, the distance to the nearest HPP located upstream or downstream is also determined. Furthermore, the data is also used to count the number of HPPs located up- and downstream of each village. To support the analysis of whether a village is located upstream or downstream, i.e., above or below a particular HPP, the altitude and other geographical features of the respective region were taken into account. An HPP is defined to be

upstream if it is closer to the origin of a river than the associated village, and downstream if it is closer to the river mouth than the associated village. To illustrate this, we use the village Thon Ninh Thanh 1 from the sample. The HPP Nong Truong 720 is located upstream of Thon Ninh Thanh 1 whereas the HPP Unknown 21 is located downstream of Thin Ninh Thanh 1 where the flow direction (from origin to mouth) is indicated by arrows (Fig 4). The MRFI did not provide names for all HPPs but instead named unknown if the name was not given.

Tables 1 and 2 give a first overview and summarize variables of interest. These tables cover information for the first and last survey waves (2007 and 2017) of the TVSEP data used for the analysis. A total of six waves are used. During data collection, enumerators always ask for data from the same reference period (May until April). The data collection of the six waves used in this study were collected in 2007, 2008, 2010, 2013, 2016, and 2017. At the beginning of the study period in 2007, 20 HPPs in the neighborhood of the study villages are observed (18 in Dak Lak, and 2 in Ha Tinh). At the end of the study period in 2017, the MRFI reports 42 HPPs (35 in Dak Lak, 4 in Ha Tinh and 3 in Thua Thien-Hue). Thus, within the 10 years of interest, the number of HPPs in each district has increased and the distance to the nearest HPP has thereby decreased. This trend is impressively visible in the province of Dak Lak.

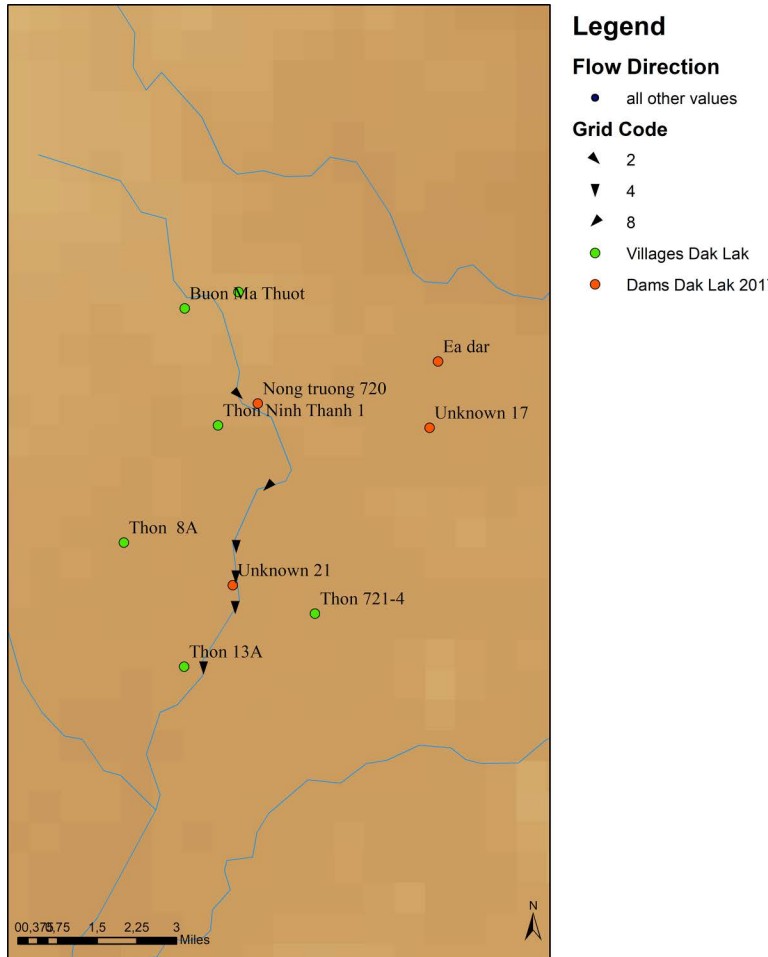

**Fig 4. Example for upstream and downstream HPPs (sourceown calculation in ArcGIS based on data from MRFI, HDX, and TVSEP, reprinted from HDX under a CC BY license, with permission from HDX, original copyright OCHA Regional Office for Asia and the Pacific (2024)).**

**Table 1. Number of small hydropower plants (HPPs) in the study sites in 2007-2017.**

|  | Dak Lak | Thua Thien-Hue | Ha Tinh |
|---|---|---|---|
| Number of HPPs 2007 | 18 | 0 | 2 |
| Number of HPPs 2017 | 35 | 3 | 4 |
| Distance to HPP 2007 | 11.72 | n.a. | 22.52 |
| Distance to HPP 2017 | 9.85 | 20.08 | 21.28 |

Source: Own calculation from TVSEP data

**Table 2. Agricultural and poverty characteristics of farm households by year.**

|  | 2007 | 2017 | T-Test |
|---|---|---|---|
| Cultivated land (ha) | 0.59 (0.93) | 0.68 (1.15) | 1.55[a] |
| Irrigated land (ha) | 0.3 (0.6) | 0.48 (0.84) | -10.59[a],*** |
| Agricultural income (2005 PPP USD) | 1,824 (3,960) | 2,229 (5,080) | -2.51[a],** |
| Expected Agric. Shocks | 4.29 (90.72) | 6.45 (7.21) | 19.03[a],*** |
| Expected Number Droughts | 3.28 (0.04) | 3.13 (0.06) | 2.16[a],** |
| Poverty headcount (village level) | 0.14 (0.15) | 0.11 (0.14) | 6.77[a],*** |
| Gini coefficient (village level) | 0.42 (0.11) | 0.36 (0.11) | 15.74[a],*** |
| No. of observations | 2,184 | 1,898 |  |

Standard deviations in parentheses; monetary values converted to 2005 PPP USD,

[a]Wilcoxon rank sum test,

*$p < 0.1$,

**$p < 0.05$,

***$p < 0.01$, source: Own calculation from TVSEP data.

Table 2 displays agricultural variables, the poverty headcount ratio, and the Gini coefficient for the entire sample rather than showing them province-wise. The area of cultivated and irrigated land increased slightly, however, while the increase in the irrigated area was significant, the increase in the cultivated land was insignificant. A possible explanation for this result is that most of the land suitable for agriculture has already been converted into cropland while farming households more and more use irrigation to better control and increase agricultural yield. Accordingly, agricultural income significantly increases during the study period. The number of agricultural shocks a farmer expects within the next year decreases significantly, which is also true if only the expected number of droughts is considered. Furthermore, the poverty headcount rate (at the village level) and the Gini coefficient (at the village level) also decrease significantly between 2007 and 2017. The descriptive data analysis suggests that the number of HPPs increases is correlated with an increase in agricultural production and a reduction in poverty.

## 3.2 Econometric modelling

The ArcGIS and TVSEP datasets were merged by exporting the ArcGIS analysis results (attribute tables) into Excel and further reading them into Stata. As not only GPS information for the villages in the TVSEP data wase added but also the village identifier, merging the data sets according to the village identifier in Stata was possible. After combining both datasets, the impact of HPPs on agricultural outcomes, poverty, and inequality was estimated using fixed effects panel regressions. Fixed effects are preferably used for this analysis, first, because the Hausman Test indicated that fixed effects produce more reliable estimates, and second

to control for time-varying factors in the whole sample and for each province individually. Thus, year-fixed effects and year-province fixed effects are included. Province fixed effects help to control for structural differences between the three provinces in the TVSEP project Furthermore, household fixed effects are also considered to account for the underlying and unobservable trends and characteristics of each household. Standard errors are clustered at the village level because the distance analysis in ArcGIS is based on the village level and, thus, the treatment due to HPPs affects each household in the village.

Two regressions are conducted separately. In the first set of regressions (equations (2a) and (3a)), the independent variable of interest consists of the distance to the closest HPP, while, in the second set of regressions (equations (2b) and (3b)), the two independent variables of interests consist of the distance to the closest upstream and downstream HPP. The aim of the first regression is to identify the effect of HPPs on agriculture and poverty of non-relocated households in the vicinity. The aim of the second regression is to differ between the impact of HPP dependent on its location regarding the villages (upstream or downstream). The dependent variables, $y_{ist}$, are agricultural income, the size of cultivated and the share of irrigated land, the number of expected droughts, the poverty headcount ratio and the Gini coefficient. While regressions (2a) and (3a) use household-level data, regressions (2b) and (3b) use village-level data. The resulting regression equations are therefore:

$$y_{ist} = \alpha + \beta DistanceHPP_{st} + X'_{ist}\Gamma + \gamma_s + \varphi_t + \vartheta_{tp} + \varepsilon_{ist} \tag{2a}$$

$$y_{ist} = \alpha + \beta DistanceDownstreamHPP_{st} + \beta DistanceUpstreamHPP_{st} + \\ X'_{ist}\Gamma + \gamma_s + \varphi_t + \vartheta_{tp} + \varepsilon_{ist} \tag{2b}$$

For the poverty headcount ratio and the Gini coefficient, which are measured at the village level, they become:

$$y_{st} = \alpha + \beta DistanceHPPVill_{st} + X'_{st}\Gamma + \gamma_s + \varphi_t + \vartheta_{tp} + \varepsilon_{st} \tag{3a}$$

$$y_{st} = \alpha + \beta NumberDownstreamHPP_{st} + \beta NumberUpstreamHPP_{st} + \\ X'_{st}\Gamma + \gamma_s + \varphi_t + \vartheta_{tp} + \varepsilon_{st} \tag{3b}$$

The other independent variables included in the regressions are represented by the vector $X'_{ist}$. It includes education, gender and age of the household head, the number of household members engaged in agriculture and in off-farm labor, and whether the household belongs to an ethnic minority. These variables were chosen due to previous studies where these variables had significant effects on agricultural outcomes such as Nguyen et al. [42]. Fixed effects are represented by $\gamma_s$ (household fixed effects), $\varphi_t$ (year fixed effects), and $\vartheta_{tp}$ (year-province fixed effects). The index $i$ indicates households, $s$ denotes villages, $p$ stands for provinces, and $t$ for years. To guarantee comparability of monetary values over time, those values are converted to 2005 US Dollar Purchasing Power Parity (PPP). The household-level independent variables are averaged at the village level for the village-level regressions.

## 4. Results and discussion

### 4.1. The impact of the closest HPP

The estimation results for equations (2a) and (3a) are presented in Table 3 (for the whole sample) and in Table 4 (only for Dak Lak). As Dak Lak has by far the highest density of HPPs (see

**Table 3. Effect of distance to nearest HPP on agriculture & poverty outcomes (whole sample).**

|  | Agricultural income | Cultivated land | Share irrigated land | Expected number droughts | Pov. head-count ratio | Gini coefficient |
|---|---|---|---|---|---|---|
| Distance to nearest HPP (km) | -10.56 | -0.025*** | -0.005** | -0.004* | 0.003 | 0.003 |
|  | (19.62) | (0.008) | (0.002) | (0.013) | (0.003) | (0.002) |
| Education household head | 18.68** | 0.006*** | 0.0006 | -0.005 | 0.0008 | 0.002 |
|  | (7.32) | (0.002) | (0.0006) | (0.003) | (0.002) | (0.002) |
| Female household head | -700.21*** | -0.14*** | -0.02 | -0.12* | -0.016 | 0.001 |
|  | (176.52) | (0.03) | (0.01) | (0.06) | (0.078) | (0.04) |
| Age household head | -8.85 | -0.003*** | -0.002** | -0.002 | -0.002 | 0.003* |
|  | (5.93) | (0.001) | (0.0003) | (0.002) | (0.003) | (0.002) |
| No. household members work on-farm | 588.82*** | 0.18*** | 0.04*** | -0.02 | 0.03* | 0.004 |
|  | (94.5) | (0.01) | (0.004) | (0.02) | (0.02) | (0.01) |
| No. household members work off-farm | -168.61*** | 0.01 | 0.0012 | -0.02 | -0.07** | 0.00 |
|  | (53.55) | (0.008) | (0.0032) | (0.02) | (0.03) | (0.01) |
| Household belongs to ethnic minority | -1,147.39 | 0.18 | -0.11*** | 0.12 | 0.13 | 0.01 |
|  | (837.98) | (0.19) | (0.03) | (0.13) | (0.22) | (0.14) |
| No. past agricultural shocks |  |  |  | 0.07*** |  |  |
|  |  |  |  | (0.023) |  |  |
| Observations | 8,558 | 9,334 | 9,331 | 4,223 | 1,163 | 1,163 |
| $R^2$ | 0.379 | 0.578 | 0.422 | 0.24 | 0.500 | 0.198 |

Standard errors clustered at village level in parentheses,

*$p < 0.1$,

**$p < 0.05$,

***$p < 0.01$, Source: Own calculation from TVSEP data.

Table 1) Dak Lak might be more suitable for the construction of HPPs due to its mountainous landscape. Furthermore, the soil in Dak Lak is suitable for cultivating water-intensive crops such as coffee and therefore has a higher need for regular water provision. This might lead to different results for the impact of HPPs and, therefore, the analysis is first conducted for the whole sample and afterwards for the restricted sample including only data from Dak Lak.

The results reveal no impact of the distance between villages and HPPs on agricultural income. At the same time, an increase in the distance to the closest HPP decreases cultivated land area by 0.025ha and the share of irrigated land area by 0.5%. The number of droughts expected in the coming year decreases by 0.005 droughts if the distance to the nearest HPP increases by one km. The results thus show that closer HPPs have a positive impact on the cultivated area and the share of irrigated land, while they have a negative impact on farmers' perceptions of the number of expected droughts. Turning to poverty and income inequality, the results show no significant impact of the distance between villages and HPPs on either, however, both tend to increase with the distance of the village to the nearest HPP. The findings are in line with the results of Duflo and Pande [4] as they also report no effects on poverty and equality indicators. Additionally, they also consider increased vulnerability to rainfall shocks to be likely as access is limited, especially in years with insufficient rainfall [21]. A further comparison with Duflo and Pande [5] reveals that the estimates for the irrigated area are similar to the results in Table 4 (0.33% vs. 0.5%). However, while Duflo and Pande [5] look at outcomes at the district level, the study at hand looks at outcomes at the household or village level. Additionally, this study uses the distance to the closest HPP, while Duflo and Pande [4] use the number of HPPs in the district. Instead of using agricultural yield as an independent

variable as in Duflo and Pande [5], this study uses agricultural income because aggregating yields from different types of crops seemed unreliable.

Higher education increases agricultural outcomes and reduces the number of expected shocks, but increases the headcount ratio and the Gini coefficient. However, the effect is only statistically significant for agricultural income and cultivated areas. The impact on agricultural outcomes is consistent with the hypotheses and evidence from previous research. A female household head significantly reduces agricultural income as well as the area under cultivation. The reduction in agricultural outcomes for female-headed households reflects their disadvantaged status and is observed in other evidence. Households with older household heads have significantly lower cultivated area and share of irrigated land. They also expect significantly fewer shocks. Villages with older household heads have, on average, a higher Gini coefficient. Older household heads may have had the opportunity to accumulate more assets and wealth than younger households. However, older household heads may be unable to be engaged intensively in agriculture due to its physically demanding activity. This may explain the results displayed in Table 4. Agricultural outcome variables are significantly higher if more household members are engaged in farming. This also significantly increases the poverty headcount, suggesting that agriculture is more strongly linked to poverty than other employment opportunities. Households with more members engaged in agriculture might face larger shocks as they are more dependent on agriculture. For household members engaged in off-farm labor, the picture is mixed. The more household members employed off-farm the lower the agricultural income. Households expect significantly more shocks when more members are engaged off-farm. Villages with households where more members work off-farm have significantly lower

**Table 4. Effect of distance to nearest HPP on agricultural & poverty outcomes (Dak Lak only).**

|  | Agricultural income | Cultivated land | Share irrigated land | Expected number droughts | Pov. headcount ratio | Gini coefficient |
|---|---|---|---|---|---|---|
| Distance to nearest HPP (km) | -79.9** | -0.03 | -0.004 | 0.025 | -0.001 | 0.003 |
|  | (35.7) | (0.02) | (0.003) | (0.016) | (0.008) | (0.006) |
| Education household head | 18.8* | 0.01*** | 0.002** | -0.005 | 0.0001 | 0.003 |
|  | (11.3) | (0.003) | (0.001) | (0.004) | (0.002) | (0.002) |
| Female household head | -1,333.04*** | -0.24*** | -0.02 | -0.07 | -0.05 | -0.05 |
|  | (424.8) | (0.06) | (0.02) | (0.09) | (0.08) | (0.06) |
| Age household head | -31.9** | -0.002 | -0.002** | -0.003 | -0.002 | -0.002 |
|  | (15.2) | (0.002) | (0.0007) | (0.003) | (0.003) | (0.003) |
| No. household members work on-farm | 877.29*** | 0.22*** | 0.03*** | -0.003 | 0.03** | 0.003 |
|  | (144.8) | (0.02) | (0.006) | (0.02) | (0.02) | (0.01) |
| No. household members work off-farm | -473.35*** | -0.02 | -0.009 | -0.07*** | -0.07*** | -0.005 |
|  | (121.2) | (0.02) | (0.006) | (0.03) | (0.02) | (0.03) |
| Household belongs to ethnic minority | -1,639.2* | 0.12 | -0.1*** | 0.12 | 0.15 | 0.2 |
|  | (942.3) | (0.23) | (0.03) | (0.13) | (0.22) | (0.17) |
| No. past agricultural shocks |  |  |  | 0.1*** |  |  |
|  |  |  |  | (0.04) |  |  |
| Observations | 3,255 | 3,495 | 3,495 | 2,095 | 1,292 | 451 |
| $R^2$ | 0.347 | 0.622 | 0.323 | 0.143 | 0.506 | 0.150 |

Standard errors clustered at village level in parentheses,

*$p < 0.1$,

**$p < 0.05$,

***$p < 0.01$, Source: Own calculation from TVSEP data.

poverty headcount ratios, again indicating that off-farm employment is superior to on-farm activities for moving out of poverty. Belonging to an ethnic minority significantly decreases the share of irrigated land but does not affect the other dependent variables. The number of agricultural shocks experienced in the past increases the number of expected droughts and is statistically significant at the 1% level.

A comparison between Table 3 and Table 4 reveals similar results for the distance between villages and HPPs. However, the effect is now significant for agricultural income, but no longer for the cultivated area and the share of irrigated land. The number of expected droughts increases by 0.025 droughts if the distance to the nearest HPP increases by one km. The distance to the nearest HPP reduces the poverty headcount ratio and increases the Gini coefficient. However, neither is statistically significant. In general, the comparison between Tables 3 and 4 shows that the coefficients are similar regarding signs but larger regarding size. This also applies to the other socioeconomic control variables included in the regression. Nevertheless, the results suggest that the three provinces respond differently to the construction of small HPPs, as the significant results appear to be driven by Thua-Thien Hue and Ha Tinh, even though the density is highest in Dak Lak. This could be due to geographical and other structural differences between these provinces, leading to different impacts from HPPs. While Dak Lak is a highland and landlocked province with a higher share of ethnic minorities, the other two provinces are coastal provinces. Duflo and Pande [5] argue that the results in districts where HPPs are located (upstream districts) tend to be noisy and insignificant. Since Dak Lak has the highest HPP density, the insignificant results might be due to this noise. Another explanation for the insignificant results in Dak Lak could be the already high density of HPPs before 2007 so adding more HPPs, even though the HPPs are closer together, might not have any effect. Consistent with the hypotheses, positive effects on agricultural outcomes emerge when small HPPs are located nearby, while the impacts of large HPPs are more diverse [5].

## 4.2. Differences in the impact of downstream and upstream HPPs

In this subsection, the effect of HPPs on the population living upstream and downstream, respectively, are estimated (equations (2b) and (3b)) and reported in Table 5 (for the whole sample) and in Table 6 (only for Dak Lak). The distance to the nearest HPP downstream determines the impact on the upstream population, while the distance to the nearest HPP upstream determines the effect on the downstream population, respectively.

Agricultural income increases for both population groups, but neither effect is significant. The cultivated area decreases for households upstream if an HPP is located nearby and increases for households downstream if an HPP is closer, but again neither effect is significant. The effect for the share of irrigated land is similar to that of cultivated area, although the effect for the downstream population is statistically significant at the 1% level. The share of irrigated land increases by 0.001ha if the distance to an HPP decreases by 1km. The number of expected droughts is not affected for the population upstream, while it decreases by 0.89 shocks when an HPP is closer to 1km for the downstream population. While the effect on the poverty headcount is insignificant for both the downstream and upstream population, the Gini coefficient increases by 0.13 if the distance to the closest HPP increases by 1 km for the upstream population. The effect on the Gini coefficient for the downstream population is not significant.

The comparison between Tables 5 and 6 shows that the number of expected droughts increases significantly with an increase in the distance between villages and HPPs for both the upstream and downstream populations. However, the effect is much larger for the population downstream. Otherwise, the results for the upstream population are similar to the results in Table 5. For the downstream population, the differences from Table 5 are more pronounced. Agricultural income decreases significantly by 65.1 USD 2005 PPP if the distance to the closest

HPP increases by 1km. As the distance to the nearest HPP downstream increases the cultivated area increases significantly, while at the same time, the share of irrigated land significantly decreases as the distance increases. The results for the further socioeconomic control variables are similar to the results from the other regression results.

When the results are compared to the existing literature and the hypotheses of this study, a closer HPP has a slight positive impact on the upstream population. However, most of the impacts on the upstream population are insignificant. These results are partly in line with the results of Duflo and Pande [5], who report insignificant but negative effects (e.g., a higher vulnerability to rainfall shocks) for the upstream population. In general, the results show more significant and positive effects of HPPs for the population downstream, namely an increase in agricultural income for Dak Lak, an increase in the share of irrigated land, and a decrease in the number of expected droughts. These findings support the hypotheses and are further substantiated by the findings of Duflo and Pande [4]. If the results are summarized and compared to those of Duflo and Pande [5], it can be concluded that the downstream population tends to fare better when small and large HPPs are nearby, while the upstream population is not affected by small HPPs nearby but is worse off when the HPPs nearby are large. Unfortunately, there are few studies that investigate non-resettled communities in terms of farming outcomes, distinguishing between upstream and downstream locations. However, the results from the Bui dam project in Ghana indicate that most farmers interviewed were not affected by the construction [43]. Still, some report land shortages because upstream households tend

**Table 5. Effect on upstream and downstream population (whole sample).**

| | Agricultural income | Cultivated land | Share irrigated land | Expected number Droughts | Pov. head-count ratio | Gini coefficient |
|---|---|---|---|---|---|---|
| Distance to nearest HPP downstream (effect on upstream households) | 1,957.1 | 0.03 | 0.05 | -0.01 | 0.15 | 0.13* |
| | (4,157.4) | (0.24) | (0.12) | (0.03) | (0.12) | (0.07) |
| Distance to nearest HPP upstream (effect on downstream households) | 31.3 | -0.009 | -0.01*** | 0.89* | -0.08 | 0.006 |
| | (43.1) | (0.02) | (0.003) | (0.45) | (0.006) | (0.004) |
| Education household head | 22.38*** | 0.005*** | 0.003 | -0.005 | 0.002 | 0.006* |
| | (8.18) | (0.002) | (0.002) | (0.004) | (0.003) | (0.003) |
| Female household head | -697.3*** | -0.16*** | -0.006 | -0.02 | -0.04 | -0.01 |
| | (156.39) | (0.03) | (0.01) | (0.09) | (0.1) | (0.08) |
| Age household head | -34.55*** | -0.002* | -0.003*** | -0.004 | -0.006 | 0.002 |
| | (16.5) | (0.001) | (0.0004) | (0.003) | (0.005) | (0.003) |
| No. household members work on-farm | 764.5*** | 0.18*** | 0.04*** | 0.003 | 0.04 | 0.009 |
| | (164.05) | (0.02) | (0.006) | (0.03) | (0.03) | (0.02) |
| No. household members work off-farm | -399.97** | 0.02* | 0.002 | 0.007 | -0.06** | -0.006 |
| | (181.1) | (0.01) | (0.004) | (0.03) | (0.03) | (0.04) |
| Household belongs to ethnic minority | -1,963.62 | 0.1 | -0.14*** | 0.06 | 0.13 | 0.28 |
| | (1,369.4) | (0.3) | (0.04) | (0.19) | (0.38) | (0.21) |
| No. past agricultural shocks | | | | 0.13** | | |
| | | | | (0.05) | | |
| Observations | 4,781 | 5,134 | 5,134 | 2,315 | 615 | 615 |
| $R^2$ | 0.135 | 0.317 | 0.405 | 0.25 | 0.504 | 0.21 |

Standard errors clustered at village level in parentheses,

*$p < 0.1$,

**$p < 0.05$,

***$p < 0.01$, Source: Own calculation from TVSEP data.

to migrate to downstream areas. This result supports the findings discussed in this chapter as it indicates that downstream farming is more profitable than upstream.

### 4.3. Robustness checks

Several robustness checks were performed in order to test the results for possible endogeneity issues. The first check investigates the fact that HPPs require certain geographical characteristics such as a river and certain gradients of the surrounding area [5]. This is followed by several robustness checks that test the robustness of the results against different definitions of dependent and independent variables. Last, the results are compared with voluntary migration and resulting selection bias.

As the placement of HPPs is not random and the villages in the area might be structurally different from other villages, lagged values of the control variables in order to solve for possible endogeneity (S2 Table) are included. The results using lagged variables are very similar to the results in Sections 4.1 and 4.2, further underpinning the robustness of the results.

To further check the robustness of the results, the number of HPPs located upstream and downstream is used instead of the distance (S3 Table). Although the results slightly change, the main conclusion remains the same. Having HPPs nearby is generally beneficial, but especially for the downstream population. This shows that the results are robust to different definitions of the effect of HPPs.

**Table 6. Effect on upstream and downstream population (Dak Lak only).**

|  | Agricultural Income | Cultivated land | Share irrigated land | Expected Number Droughts | Pov. Head-count Ratio | Gini Coefficient |
|---|---|---|---|---|---|---|
| Distance to nearest HPP downstream (effect on upstream households) | -1,943.8 | 0.03 | 0.05 | 0.03* | 0.14 | 0.13* |
|  | (4,169.4) | (0.24) | (0.12) | (0.02) | (0.12) | (0.07) |
| Distance to nearest HPP upstream (effect on downstream households) | -65.1* | 0.013* | -0.011*** | 0.89* | -0.002 | -0.003 |
|  | (35.5) | (0.007) | (0.003) | (0.45) | (0.005) | (0.003) |
| Education household head | 44.05*** | 0.01*** | 0.001 | 0.0004 | 0.005 | 0.006* |
|  | (15.8) | (0.004) | (0.002) | (0.005) | (0.006) | (0.003) |
| Female household head | -1,124.9** | -0.28*** | -0.01 | 0.02 | 0.05 | -0.04 |
|  | (441.9) | (0.08) | (0.02) | (0.15) | (0.18) | (0.13) |
| Age household head | -36.16* | -0.0001 | -0.004*** | -0.005 | -0.003 | -0.003 |
|  | (18.69) | (0.004) | (0.001) | (0.005) | (0.009) | (0.004) |
| No. household members work on-farm | 862.64*** | 0.23*** | 0.04*** | 0.06 | 0.03 | 0.017 |
|  | (240.3) | (0.04) | (0.009) | (0.03) | (0.04) | (0.016) |
| No. household members work off-farm | -486.21*** | -0.026 | -0.01 | -0.009 | -0.24*** | -0.006 |
|  | (201.9) | (0.02) | (0.01) | (0.04) | (0.08) | (0.04) |
| Household belongs to ethnic minority | -2,514.8 | 0.1 | -0.13*** | 0.07 | 0.12 | 0.47* |
|  | (1,535.8) | (0.3) | (0.04) | (0.18) | (0.4) | (0.28) |
| No. past agricultural shocks |  |  |  | 0.13** |  |  |
|  |  |  |  | (0.06) |  |  |
| Observations | 1,573 | 1,376 | 1,376 | 1,008 | 225 | 225 |
| $R^2$ | 0.134 | 0.269 | 0.247 | 0.183 | 0.427 | 0.25 |

Standard errors clustered at village level in parentheses,

$^*p < 0.1$,

$^{**}p < 0.05$,

$^{***}p < 0.01$, Source: Own calculation from TVSEP data.

Furthermore, varying agricultural variables by using cash crop production instead of agricultural income (e.g., coffee, tobacco and tea) as those crops need more irrigation and their production might, thus, be affected differently by HPPs in comparison to other crops. First, cash crop production is used as the dependent variable. The results show a mainly insignificant effect of HPPs on the production of cash crops (S4 Table). Second, a dummy for whether a household grows cash crops is added as a control variable to equations 1a & b, and 2a & b. However, the effect of the cash crop dummy mainly has a significant impact on the share of irrigated land, as these crops require more intensive irrigation (S5 Table). However, the coefficients for the HPPs change only marginally in magnitude, while the statistical significance remains stable. In addition, stable irrigation due to close HPPs might incentivize farmers to grow cash crops. Therefore, an interaction term between the HPP variables and an indicator variable for the growth of cash crops is added as an additional control variable. Although the coefficient for the interaction term is statistically significant for some specifications, the results for the HPPs themselves are robust (S6 Table).

To check the robustness of the results against the definition of weather shocks, the number of expected agricultural shocks is used instead of the number of expected droughts, which also include landslides, unusually heavy rainfall and floods (S7 Table). The results are quite similar to those for the number of expected droughts.

Due to the construction of HPPs disadvantaged households might migrate and thus drop out of future data collection for the TVSEP project. There is a concern that this introduces selection bias as households negatively affected by HPP construction choose to migrate and the dataset only includes households that benefit or are not affected by HPPs. Therefore, a balanced sample is created and compared with the sample of households for which data for all six waves are not available. To test for differences between these two samples, t-tests are calculated. S8 Table shows significant differences between the two groups and reveals that households that leave the sample are worse off than households that remain in the area. For example, such households are more likely to have a female household head, are poorer, irrigate smaller parts of their cropland, have less agricultural production, and have fewer household members engaged in farming. However, the unbalanced sample is much smaller than the balanced sample. Additionally, they are surrounded by fewer upstream and downstream HPPs and are generally farther away from the nearest HPP. To further test for selection bias, the nucleus household size is added as a dependent variable to test whether migration increases with the construction of HPPs (S9 Table). The results for Dak Lak show that HPPs have no impact on the size of nucleus households for the upstream population, but the opposite for downstream households. One reason for this can be the generally positive effect of HPPs on the population downstream. In conclusion, distinguishing between households remaining in the sample and households leaving the samples reveals structural differences between the two groups. However, it is not clear whether this is due to the construction of HPPs or to other household-specific characteristics. The structure of the present dataset does not allow for further investigation and therefore this relationship deserves detailed analysis and opens future research opportunities.

Lastly, former findings suggest an effect of HPP construction on the productivity of land [23,24]. Accordingly, the average yield per hectare is used as the dependent variable. The results, however, are mostly insignificant (S10 Table).

## 5. Conclusion

Vietnam has experienced a significant growth in small HPPs demanding research into their effects on nearby farming households. While previous research, often relying on case studies,

has focused on large HPPs and resettled households [6,11,17,18], small HPPs might have different impacts. This study examines small HPPs' effect on non-relocated households' well-being, using fixed effects regression on extensive panel data from three Vietnamese provinces. First, the effects of the proximity of HPPs to villages on agricultural income and cultivated land are analyzed. Findings indicate that nearby HPPs increase agricultural income and irrigated land while reducing the number of expected droughts. Thus, stable irrigation from HPPs benefits agriculture, but does not impact poverty or equality. These findings align with former research results [5,11] and support the hypotheses. Second, the investigation differentiates effects based on village location relative to HPPs. Results show downstream villages benefit from increased irrigated areas, while upstream villages remain unaffected, suggesting no adverse effects on upstream populations. Small HPPs prevent negative impacts associated with large HPPs, like relocation and reduced agricultural yields, offering advantages such as minimal household relocation and potentially lesser effects on land quality and aquatic life. Environmental impacts require further interdisciplinary research.

Small HPPs offer benefits comparable to large HPPs and should be encouraged. They positively impact agriculture with potentially fewer negative effects on upstream populations. Therefore, policymakers should prioritize small HPP construction over large ones. Future research could explore the cumulative effects of multiple small HPPs built on the same river on nearby communities. Additionally, it is crucial for policymakers to recognize the diverging effects on upstream and downstream populations. Preventing social conflicts requires adequate compensation, clear communication, and involving affected villages in the planning process.

Future research should explore the environmental and job creation impacts of small versus large HPPs, as this study primarily focuses on agricultural outcomes. While large HPPs are known to harm soil fertility, increase water salinity and negatively affect fish populations, the environmental effects of small HPPs remain insufficiently examined. Investigating these impacts and their influence on agriculture and well-being requires interdisciplinary research, including natural science. Furthermore, the relationship between HPPs and migration needs further examination. While forced displacement is well-researched [16,21,44], it is unclear how HPPs affect migration flows. People may migrate closer for perceived benefits like job opportunities [5] or move away due to anticipated negative impacts. Robustness checks indicate a possible selection bias due to significant differences between households moving away and those remaining in the study area. Combining TVSEP data with TVSEP migration data could address this issue, by tracking migrated households. Moreover, the average 10 km distance between HPPs and villages might affect the results. Future research should identify and collect targeted data on households in very proximity to HPPs to facilitate comparison with those further away. This effort would enhance understanding of HPPs' comprehensive impact on communities.

## Supporting information

**S1 Table. Agricultural variables calculated for each household.**
(DOCX)

**S2 Table. Fixed effects regression for the impact of HPPs using lagged values for control variables.**
(DOCX)

**S3 Table. Results for equations (2b) & (3b) using numbers of HPPs upstream and downstream.**
(DOCX)

**S4 Table. HPPs' effect on cash crop production.**
(DOCX)

**S5 Table. Distance to nearest HPP with cash crop dummy as additional control variable.**
(DOCX)

**S6 Table. Distance to nearest HPP with cash crop dummy and interaction terms as additional control variable.**
(DOCX)

**S7 Table. HPPs' effect on expected number of agricultural shocks for equations (2a) & (2b).**
(DOCX)

**S8 Table. Test for differences in means for households who dropped out and stayed in the sample.**
(DOCX)

**S9 Table. HPPs' effect on household size using household size for equations (2a) & (2b).**
(DOCX)

**S10 Table. HPPs' effect on average yield for equations (2a) & (2b).**
(DOCX)

## Acknowledgements

This study relies on data from the long-term project No. 20220831434900116103. For more detailed information, see http://www.tvsep.de. We greatly acknowledge the constructive comments from two anonymous reviewers.

## Author contributions

**Conceptualization:** Eva Seewald, Trung Thanh Nguyen.

**Formal analysis:** Eva Seewald.

**Methodology:** Eva Seewald, Ulrike Grote, Trung Thanh Nguyen.

**Supervision:** Ulrike Grote, Trung Thanh Nguyen.

**Writing – original draft:** Eva Seewald.

**Writing – review & editing:** Eva Seewald, Ulrike Grote, Trung Thanh Nguyen.

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
