## [Decision Letter · Decision Letter 0]

11 Oct 2024

PONE-D-24-38432Small hydropower plants and livelihoods of the local population in rural VietnamPLOS ONE

Dear Dr. Nguyen,

Thank you for submitting your manuscript to PLOS ONE. After careful consideration, we feel that it has merit but does not fully meet PLOS ONE’s publication criteria as it currently stands. Therefore, we invite you to submit a revised version of the manuscript that addresses the points raised during the review process.

**ACADEMIC EDITOR COMMENTS: **

The study is interesting and the obtained results may useful for further activities or applications in the specified area. However, the manuscript preparation is not well. The manuscript organization should strictly adhere to the authors guidelines of the journal.  Accordingly, authors should revise the manuscript under the headings of Introduction, Materials and Methods, Results, and Discussion sections only.

The introduction is too vague and elaborated. Study background information should be very clear and more specific to the study objectives. Please remove all subheadings, figures, equations and broad literature review in the introduction section. Authors should minimize the introduction section.

Please revise the other sections strictly with the headings “Materials and Methods, Results, and Discussion sections and avoid too general statements.

Study conclusions are not clear in the manuscript. Please revise the conclusion section with study highlights and findings significance. Remove elaborated descriptions from the conclusion.

We look forward to receiving your revised manuscript.

Kind regards,

S Ezhil Vendan, Ph.D

Academic Editor

PLOS ONE

“German Research Foundation”

“This study relies on data from the long-term project No. 20220831434900116103, funded by the Deutsche Forschungsgemeinschaft (DFG). For more detailed information, see http://www.tvsep.de”

“German Research Foundation”

5. We note that Figures to 2,3 and 4 in your submission contain [map/satellite] images which may be copyrighted. All PLOS content is published under the Creative Commons Attribution License (CC BY 4.0), which means that the manuscript, images, and Supporting Information files will be freely available online, and any third party is permitted to access, download, copy, distribute, and use these materials in any way, even commercially, with proper attribution. For these reasons, we cannot publish previously copyrighted maps or satellite images created using proprietary data, such as Google software (Google Maps, Street View, and Earth). For more information, see our copyright guidelines: http://journals.plos.org/plosone/s/licenses-and-copyright.

a. You may seek permission from the original copyright holder of Figures to 2,3 and 4 to publish the content specifically under the CC BY 4.0 license. 

Additional Editor Comments:

The study is interesting and the obtained results may useful for further activities or applications in the specified area. However, the manuscript preparation is not well. The manuscript organization should strictly adhere to the authors guidelines of the journal. Accordingly, authors should revise the manuscript under the headings of Introduction, Materials and Methods, Results, and Discussion sections only.

The introduction is too vague and elaborated. Study background information should be very clear and more specific to the study objectives. Please remove all subheadings, figures, equations and broad literature review in the introduction section. Authors should minimize the introduction section.

Please revise the other sections strictly with the headings “Materials and Methods, Results, and Discussion sections and avoid too general statements.

Study conclusions are not clear in the manuscript. Please revise the conclusion section with study highlights and findings significance. Remove elaborated descriptions from the conclusion.

Reviewers' comments:

Reviewer's Responses to Questions

**Comments to the Author**

1. Is the manuscript technically sound, and do the data support the conclusions?

Reviewer #1: Yes

Reviewer #2: Partly

2. Has the statistical analysis been performed appropriately and rigorously? 

Reviewer #1: I Don't Know

Reviewer #2: Yes

3. Have the authors made all data underlying the findings in their manuscript fully available?

Reviewer #1: Yes

Reviewer #2: Yes

4. Is the manuscript presented in an intelligible fashion and written in standard English?

Reviewer #1: Yes

Reviewer #2: No

5. Review Comments to the Author

Reviewer #1: Dear Author,

Thank you for submitting your manuscript. Your study on the impact of small hydropower plants (HPPs) on rural livelihoods in Vietnam provides valuable insights into an important and under-researched area. By leveraging a large panel dataset and focusing on the effects of small HPPs, your research addresses critical gaps in the current literature.

To enhance the overall clarity and effectiveness of your manuscript, I have outlined a set of observations and suggestions relevant to various sections of your study, including the introduction, methodology, results, and conclusions. These recommendations are intended to refine and strengthen your work. Please review the following points to improve your manuscript.

I suggest change the word hydropower in Keywords: Hydropower because is on the title.

Use third-person perspective throughout the text to maintain a formal and objective tone.

Intoduction

[1] The motivation could be strengthened by providing more specific examples or recent data on the impact of HPPs, particularly focusing on small HPPs, to highlight the current relevance of the research.

[2] It would be beneficial to explicitly state how the study's findings could influence policy or practical applications. Additionally, briefly outlining the main findings in the introduction could further emphasize the study's contribution.

Data and Methodoloy

[3]The description is adequate but could be improved by specifying the exact timeframe covered by the TVSEP dataset. Although you mention the use of data from 2007 to 2017, providing specific dates of data collection, if available, would be beneficial.

[4] The description of the MRFI dataset is clear. It would be helpful to mention if the dataset is regularly updated and how current the data are. This will support the temporal validity of your results.

[5] The methodology for merging data using GPS and ArcGIS is well-explained. You might enhance this section by briefly describing how data quality was ensured during the merging process and the accuracy of GPS coordinates. This will bolster confidence in your results.

[6] In the section about using ArcGIS, you mention that results are transferred to Stata for analysis. Adding details on how data conversion was managed and if there were any significant challenges would be beneficial.

[7] The details on agricultural variables are good, but you could add specific examples of how variables like "agricultural shocks" and "expected number of droughts" were measured. Clarifying these points will improve transparency and reproducibility.

[8] When calculating poverty indices and Gini coefficients, it would be helpful to provide more information on the poverty threshold used and the definition. This will contextualize the results.

[9] The tables provide a useful overview, but a more detailed analysis of observed trends would be beneficial. For instance, explaining why changes in cultivated land area and agricultural income are significant or not can offer deeper insights into the results.

[10] The explanation of regression models is clear but could be improved by detailing how control variables were selected and why they are relevant to the study. Additionally, briefly explaining why fixed effects models were chosen and how they help control for unobserved variables would add rigor.

[11] Providing justification for using cluster-robust standard errors at the village level would also strengthen the methodology.

Results and Discussion

[12] Clearly explain why results differ between the overall sample and Dak Lak.

[13] Provide more context on how socio-economic variables like education, gender, and age of household heads influence outcomes related to HPPs.

[14]Compare findings more thoroughly with previous studies, such as Duflo and Pande (2007), highlighting both similarities and differences.

[15] Discuss why some results are significant in one province but not in others, and the possible reasons for insignificant results.

[16] Expand on the robustness of the results by detailing how robustness checks address potential endogeneity issues and other concerns.

[17] Analyze in more detail how HPPs impact upstream and downstream populations differently, including specific examples or case studies.

[18] Improve the visual presentation of data, such as using clearer graphs or tables, and ensure they are well-labeled and easy to interpret.

[19] Discuss selection bias more explicitly, including how it was addressed and its potential impact on the results.

Identify specific areas for future research based on current findings and limitations, and suggest additional methodological approaches.

Conclusion

[20] Clearly articulate how the findings on small HPPs compare with those on large HPPs, beyond noting that small HPPs show no significant negative effects on upstream populations.

[21]Provide a more detailed explanation of the specific environmental impacts of small HPPs and compare these impacts with those of large HPPs.

[22]Discuss the implications of the observed positive effects of small HPPs on downstream populations in the context of their overall benefit compared to large HPPs.

[23]Elaborate on how the results might influence policy recommendations for promoting small HPPs, including specific measures to mitigate potential negative impacts.

[24] Address the gap in understanding regarding the relationship between HPPs and migration, and suggest how future research could explore this aspect in more detail.

[25] Highlight any limitations of the current study and how they might affect the interpretation of results, including the scope of the panel dataset and potential biases.

Reviewer #2: General comments

This manuscript small hydropower plants and livelihoods of the local population in rural Vietnam. There is a lot to like about the research presented in this article. Therefore, the following comments intend to support the authors to ensure the proper delivery of the research presented in this article:

1- Introduction

• The introduction lacks a clear, concise topic sentence for each paragraph, leading to unclear transitions between thoughts. The authors should introduce key ideas more effectively to improve coherence.

• The introduction does not include hypotheses or research questions, which are crucial in guiding the reader through the study's aim. The authors should include these to clarify the direction of the research.

• The introduction lacks a detailed literature review that situates the study within the existing body of research. Including a review of relevant literature would provide context and justify the need for this research.

• The introduction does not provide information about the current state of agriculture in Vitamin.

2- Methodology

The current Research Design sub-section does not explain the design of this research. Instead, the current Section 2.2 contains Data Collection (please rename it). Therefore, the authors should make a Research Design sub-section to explain the step-by-step (stages) of this research. It is critical to convince readers that this research was conducted systematically and included necessary research activities. For each stage, please explain its objective(s), technique/approach(es) being used, and outcome(s) expected from the stage.

3- Results

Results and Discussion do not get along. The authors should focus on presenting the results of this study in this Results section. The authors do not compare their findings with those of other studies in similar or different contexts. A comparative discussion, especially with findings from studies in different regions or farming systems, would provide a more comprehensive understanding of how generalizable or context-specific the results are

4- conclusion

The current form of Conclusion mixes up the summary and key findings of this study with suggestions arising from the findings. The authors may simply consider using 3 paragraphs:

i. Summary and key findings of this study

ii. Managerial and policy implications arising for each key finding

iii. Insights for future research arising from the key findings and limitations of this research

6. PLOS authors have the option to publish the peer review history of their article (what does this mean? ). If published, this will include your full peer review and any attached files.

**Do you want your identity to be public for this peer review?** For information about this choice, including consent withdrawal, please see our Privacy Policy .

Reviewer #1: **Yes: ** Carlos Alberto Zuniga Gonzalez

Reviewer #2: No

---

## [Author Response · Author response to Decision Letter 1]

14 Nov 2024

We have addressed all comments from the two reviewers. Our responses are submitted separately.

---

## [Decision Letter · Decision Letter 1]

13 Dec 2024

PONE-D-24-38432R1Small hydropower plants and livelihoods of the local population in rural VietnamPLOS ONE

Dear Dr. Nguyen,

Thank you for submitting your manuscript to PLOS ONE. After careful consideration, we feel that it has merit but does not fully meet PLOS ONE’s publication criteria as it currently stands. Therefore, we invite you to submit a revised version of the manuscript that addresses the points raised during the review process.

**ACADEMIC EDITOR: **

The revised manuscript is satisfactory. However, the manuscript needs revision.

We look forward to receiving your revised manuscript.

Kind regards,

S Ezhil Vendan, Ph.D

Academic Editor

PLOS ONE

Journal Requirements:

**Additional Editor Comments:**

The revised manuscript is satisfactory. However, the manuscript needs minor revision.

Page 9: Please check "(Error! Reference source not found.)" and revise.

In conclusion: The conclusion section was elaborately described. Avoid detailed background information and discussions in conclusion section. Please check and reduce the conclusion with respect to highlights of the obtained results.

Add sources of maps in the methodology sections.

Please check the typographical errors through out the manuscript.

Please check the references and revise as per the journal guidelines. For example, World bank reference in the reference list, page numbers uniformity in all the references (e.g., 2,4,5,8, etc.,), check the reference format in reference 1,13,14,15,18,19, etc. (add website link).

Reviewers' comments:

Reviewer's Responses to Questions

**Comments to the Author**

1. If the authors have adequately addressed your comments raised in a previous round of review and you feel that this manuscript is now acceptable for publication, you may indicate that here to bypass the “Comments to the Author” section, enter your conflict of interest statement in the “Confidential to Editor” section, and submit your "Accept" recommendation.

Reviewer #1: All comments have been addressed

2. Is the manuscript technically sound, and do the data support the conclusions?

Reviewer #1: Yes

3. Has the statistical analysis been performed appropriately and rigorously? 

Reviewer #1: Yes

4. Have the authors made all data underlying the findings in their manuscript fully available?

Reviewer #1: Yes

5. Is the manuscript presented in an intelligible fashion and written in standard English?

Reviewer #1: Yes

6. Review Comments to the Author

Reviewer #1: After analyzing the documents, it is evident that the authors systematically addressed the reviewers' comments. Here's a summary of their adherence to the reviewers' feedback:

Structural Revisions:

The manuscript was reorganized to adhere to the journal's format, maintaining combined Results and Discussion sections due to readability benefits. The Conclusion section was significantly improved, structured into three clear parts as suggested: summary, policy implications, and future research.

Introduction Improvements:

The authors clarified the study’s aim and added hypotheses as requested. Background information on Vietnam's agricultural sector was included, and the section was streamlined by moving the literature review to a dedicated section.

Methodological Adjustments:

They clarified the rationale for their chosen regression models, selection of control variables, and addressed the suggestion for more detailed explanations on data sources and merging processes.

Results and Comparative Analysis:

Comparisons with other studies and specific regional insights (e.g., Dak Lak province) were added to enhance the interpretation of findings, addressing concerns about generalizability.

Conclusion Refinement:

Suggestions regarding key findings, policy recommendations, and limitations were incorporated. The discussion of limitations was expanded to highlight gaps and biases.

Addressing Specific Reviewer Concerns:

Reviewer-specific issues, such as language tone, motivation strengthening, and removing redundant keywords, were fully addressed. Suggestions to enhance figure copyrights and data-sharing compliance were also implemented.

Conclusion:

The authors appear to have incorporated all major comments, based on their detailed responses and manuscript revisions.

7. PLOS authors have the option to publish the peer review history of their article (what does this mean? ). If published, this will include your full peer review and any attached files.

**Do you want your identity to be public for this peer review?** For information about this choice, including consent withdrawal, please see our Privacy Policy .

Reviewer #1: No

---

## [Author Response · Author response to Decision Letter 2]

19 Dec 2024

Responses to the Academic Editor

1. Your comment: Page 9: Please check "(Error! Reference source not found.)" and revise.

Our response: We are sorry for this. It has been corrected.

2. Your comment: In conclusion: The conclusion section was elaborately described. Avoid detailed background information and discussions in conclusion section. Please check and reduce the conclusion with respect to highlights of the obtained results.

Our response: Thank you for making our manuscript easier to understand. We shortened the conclusion. The conclusion now reads:

“Vietnam has experienced a significant growth in small HPPs demanding research into their effects on nearby farming households. While previous research, often relying on case studies, has focused on large HPPs and resettled households [6,11,17,18], small HPPs might have different impacts. This study examines small HPPs’ effect on non-relocated households’ well-being, using fixed effects regression on extensive panel data from three Vietnamese provinces. First, the effects of the proximity of HPPs to villages on agricultural income and cultivated land are analyzed. Findings indicate that nearby HPPs increase agricultural income and irrigated land while reducing the number of expected droughts. Thus, stable irrigation from HPPs benefits agricultural, but does not impact poverty or equality. These findings align with former research results [5,11] and support the hypotheses. Second, the investigation differentiates effects based on village location relative to HPPs. Results show downstream villages benefit from increased irrigated areas, while upstream villages remain unaffected, suggesting no adverse effects on upstream populations. Small HPPs prevent negative impacts associated with large HPPs, like relocation and reduced agricultural yields, offering advantages such as minimal household relocation and potentially lesser effects on land quality and aquatic life. Environmental impacts require further inter-disciplinary research.

Small HPPs offer benefits comparable to large HPPs and should be encouraged. They positively impact agriculture with potentially fewer negative effects on upstream populations. Therefore, policymakers should prioritize small HPP construction over large ones. Future research could explore the cumulative effects of multiple small HPPs built on the same river on nearby communities. Additionally, it is crucial for policymakers to recognize the diverging effects on upstream and downstream populations. Preventing social conflicts requires adequate compensation, clear communication, and involving affected villages in the planning process.

Future research should explore the environmental and job creation impacts of small versus large HPPs, as this study primarily focuses on agricultural outcomes. While large HPPs are known to harm soil fertility, increase water salinity and negatively affect fish populations, the environmental effects of small HPPs remain insufficiently examined. Investigating these impacts and their influence on agriculture and well-being requires interdisciplinary research, including natural science. Furthermore, the relationship between HPPs and migration needs further examination. While forced displacement is well-researched [16,21,44], it is unclear how HPPs affect migration flows. People may migrate closer for perceived benefits like job opportunities [5] or move away due to anticipated negative impacts. Robustness checks indicate a possible selection bias due to significant differences between households moving away and those remaining in the study area. Combining TVSEP data with TVSEP migration data could address this issue, by tracking migrated households. Moreover, the average 10 km distance between HPPs and villages might affect the results. Future research should identify and collect targeted data on households in very proximity to HPPs to facilitate comparison with those further away. This effort would enhance understanding of HPPs’ comprehensive impact on communities.”

3. Your comment: Add sources of maps in the methodology sections.

Our response: We added the data sources to the maps

4. Your comment: Please check the typographical errors throughout the manuscript.

Our response: We checked the manuscript for errors and revised it accordingly.

5. Your comment: Please check the references and revise as per the journal guidelines. For example, World bank reference in the reference list, page numbers uniformity in all the references (e.g., 2,4,5,8, etc.,), check the reference format in reference 1,13,14,15,18,19, etc. (add website link).

Our response: We checked the references, added information and revised according to the journal guidelines.

Once again, we would like to thank you very much for your comments.

---

## [Editor Report · Decision Letter 2]

26 Dec 2024

Small hydropower plants and livelihoods of the local population in rural Vietnam

PONE-D-24-38432R2

Dear Dr. Trung Thanh Nguyen,

We’re pleased to inform you that your manuscript has been judged scientifically suitable for publication and will be formally accepted for publication once it meets all outstanding technical requirements.

Kind regards,

S Ezhil Vendan, Ph.D

Academic Editor

PLOS ONE

---

## [Editor Report · Acceptance letter]

PONE-D-24-38432R2

PLOS ONE

Dear Dr. Nguyen,

I'm pleased to inform you that your manuscript has been deemed suitable for publication in PLOS ONE. Congratulations! Your manuscript is now being handed over to our production team.

Kind regards,

on behalf of

Dr. S Ezhil Vendan

Academic Editor

PLOS ONE